# DFED: Data-Free Ensemble Distillation with Multi-Source GANs for Heterogeneous Federated Learning

## Abstract

Federated Learning (FL) is a decentralized machine learning paradigm that enables clients to collaboratively train models while preserving data privacy. However, surmounting the obstacles introduced by data heterogeneity in heterogeneous federated learning remains a profound challenge, as it drives each client towards distinct convergence trajectories, impeding the global model's convergence. To transcend these challenges, we propose DFED, a novel data-free ensemble knowledge distillation method designed to counteract the effects of data heterogeneity. DFED leverages multi-source Generative Adversarial Networks (GANs) to generate synthetic data that aligns with local distributions, ensuring privacy while promoting diverse feature representations across clients. Additionally, DFED aggregates client models into an ensemble based on their specialized knowledge, and applies ensemble distillation to refine the global model, mitigating the issues caused by disparities in data distributions. Across a variety of image classification benchmarks, DFED demonstrates superior performance compared to several state-of-the-art (SOTA) methods. The source code will be made publicly accessible once the paper has been accepted for publication.

## 1 Introduction

Federated Learning (FL) has emerged as a pivotal paradigm in the realm of machine learning, driven by the increasing demand for privacy-preserving computational frameworks (Yang et al., 2019; Aledhari et al., 2020a). In contrast to traditional centralized learning, FL enables multiple clients, each possessing their own local datasets, to collaboratively train a global model without the need to exchange raw data (Li et al., 2023). This methodology not only facilitates effective collaboration but also strengthens privacy protection by eliminating the direct transfer of sensitive information, thereby significantly reducing the risks of data leakage and unauthorized access(Matsuda et al., 2021; Shi et al., 2024).

However, the promise of Federated Learning does not come without significant challenges, chief among them being data heterogeneity (Li et al., 2020; Konecny et al., 2015; Ye et al., 2023; Mendieta et al., 2022). In practical scenarios, data possessed by different clients can vary significantly due to differences in user behavior, local environments, or underlying data-generating processes(Zhang et al., 2021). This variation in data, typically characterized as non-IID (Independent and Identically Distributed), further exacerbates the difficulty in achieving a uniformly performing global model(Aledhari et al., 2020b; Zhao et al., 2018; Zhu et al., 2021a). Specifically, when clients possess heterogeneous data, their local models tend to diverge during training, adapting to the distinct characteristics of their respective datasets. This divergence, known as client drift(Karimireddy et al., 2020), leads to models that reflect the disparities of private data rather than contributing towards a unified global objective. As a result, the trained global model may perform well

on some clients' data but struggle to generalize effectively to others, causing inconsistent performance and reduced fairness across clients (Shang et al., 2022). Directly aggregating model parameters or updates in such scenarios can further reduce the global model's overall performance, leading to fairness concerns and diminished transferability.

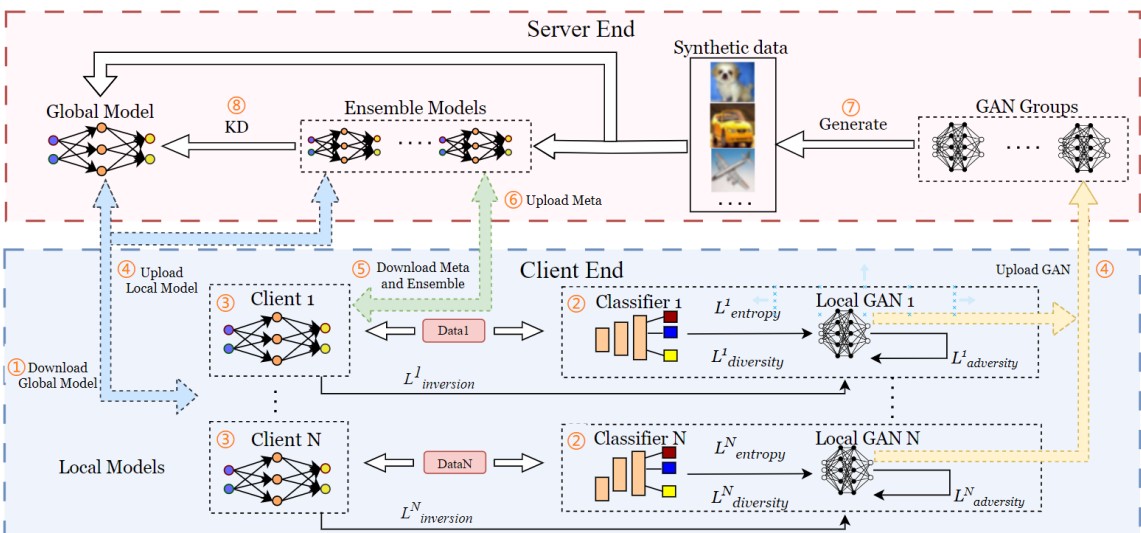

Figure 1: Overview of the federated learning framework with multi-source GANs for data-free ensemble distillation. In the general phase, represented by ①–④, the global model is distributed, local GANs and models are trained and uploaded. In the meta phase, shown by ⑤–⑥, the ensemble models and meta-head are trained across selected clients, leveraging EMA to prevent forgetting. After the meta phase, knowledge distillation ⑦–⑧ is performed using the synthetic data generated by the GANs to improve the global model.

With the progression of Federated Learning (FL), Knowledge Distillation (KD)(Hinton et al., 2015) has emerged as a pivotal technique for transferring knowledge from a large, complex model (teacher) to a smaller, more efficient model (student)(Gou et al., 2021; Wu et al., 2021). Widely applied in tasks such as model compression, transfer learning, and domain adaptation, KD enables the student model to assimilate the teacher's knowledge with minimal performance degradation(Park et al., 2019). This not only simplifies model complexity but also enhances adaptability and robustness, particularly in settings characterized by diverse data distributions. Unlike conventional FL approaches that aggregate model weights—often exacerbating heterogeneity—KD facilitates learning from a distilled global representation, allowing client models to better align with their local data and architecture(Zhang et al., 2024b; Qiao et al., 2023). For instance, the works of Jiang et al. (2020) and Ma et al. (2022) illustrate how knowledge distillation can enhance federated learning by efficiently transferring knowledge from local models and mitigating catastrophic forgetting, thereby improving the global model's performance across heterogeneous and continual learning scenarios. Nevertheless, methods such as FedMD(Li & Wang, 2019), which depend on publicly available datasets for distillation, present challenges in privacy-sensitive contexts due to the risk of exposing sensitive client information.

To address these limitations, data-free knowledge distillation (DFKD) has emerged as a promising alternative(Lopes et al., 2017; Luo et al., 2020; Liu et al., 2024). By eliminating the dependency on public datasets, DFKD ensures that sensitive client data remains protected while still allowing the global model to leverage the knowledge of individual clients(Zhu et al., 2021b). Building upon the framework of DFKD, we propose a novel approach called DFED to address data heterogeneity and privacy concerns in federated

learning by integrating ensemble knowledge distillation with Generative Adversarial Networks (GANs) for synthetic data generation. Firstly, to safeguard data privacy, we deploy GANs on each client to generate synthetic data reflective of their local distributions. These GANs are subsequently integrated into a unified collection on the server, offering valuable and diverse samples for the knowledge distillation process. Subsequently, to mitigate the inherent Non-IID nature of the data—which restricts local models to excel in only distinct tasks—we aggregate the local models into a specialized ensemble, with each model focusing on particular objectives, leading to a substantial improvement in predictive performance compared to the global model alone. Lastly, we refine this integration through attention-based meta-learning, followed by knowledge distillation, wherein the model ensemble serves as the teacher and the global model as the student. This three-step methodology ensures iterative enhancement of the global model's performance.

Our primary contributions are summarized as follows. First, we introduce an innovative federated learning method that enhances the model's effectiveness in heterogeneous environments. Second, we explore the use of GANs in scenarios characterized by data imbalance, where each client trains its own GAN. The collective deployment of these GANs generates diverse synthetic data, ensuring both distribution uniqueness and privacy preservation. Moreover, we leverage a combination of model ensembles and attention-based meta-learning to significantly elevate the performance of the ensemble beyond that of a conventional global model. Finally, we utilize knowledge distillation with the generated synthetic data alongside the high-performing model ensemble, resulting in further performance improvements. Our approach demonstrates significant superiority over several state-of-the-art methods on the CIFAR-10 and CIFAR-100 datasets.

## 2 RELATED WORK

Due to space limitations, this part have been moved to Appendix A.1.

## 3 PROPOSED METHOD

In this section, we first introduce some basic notations and then provide a detailed explanation of the proposed method DFED. We consider DFED as a optimization technique specifically designed to address the challenges posed by data heterogeneity in federated learning. The framework of DFED is depicted in Fig. 1, illustrating its key components and workflow.

### 3.1 PRELIMINARIES

**Notations.** In this paper, we consider a classical federated learning setup with $N$ clients, each owning private labeled datasets $\{(X_i, Y_i)\}_{i=1}^N$, where $X_i = \{x_i^b\}_{b=1}^{n_i}$ follows the data distribution $D_i$ over feature space $\mathcal{X}_i$, i.e., $x_i^b \sim D_i$. These clients collaborate on a classification task with $C$ classes, where $Y_i = \{y_i^b\}_{b=1}^{n_i} \subset \{1, \ldots, C\}$ represents the ground-truth labels corresponding to the samples in $X_i$. Notably, We focus only on the issue of data heterogeneity. Specifically, while the feature space remains the same for all clients, the data distributions may differ across clients. This manifests as label distribution skewness among clients, i.e., $\mathcal{X}_i = \mathcal{X}_j$ and $D_i \neq D_j$, $\forall i \neq j, i, j \in [N]$.

The batch size used for local training is represented by $B$, the weight matrix of the final classification layer is denoted by $W = [w_1, w_2, \ldots, w_C]^\top \in \mathbb{R}^{C \times d}$, and for simplicity, bias terms are omitted. Our objective is to train a global model without requiring the clients to upload their data to the central server. The objective of the global model optimization can be formulated as minimizing the following loss function:

$$\min_\theta \sum_{i=1}^N \frac{|D_i|}{|D_{\text{total}}|} \mathcal{L}_i(F_\theta(D_i), Y_i)$$

where $\theta$ represents the parameters of the global model, $\mathcal{L}_i$ denotes the local loss function for client $i$, and $|D_{\text{total}}| = \sum_{i=1}^{N} |D_i|$ is the total size of datasets across all clients.

**Basic Algorithm of Federated Learning**. We use FedAvg (McMahan et al., 2016) as the core algorithm. The standard federated learning process follows these steps: In round $t$, the server distributes the global model $\mathbf{w}^t$ to all participating clients. Each client $k$, based on its local dataset $D_k$, updates the local model $\mathbf{w}_k^t$ using the following rule:

$$\mathbf{w}_k^{t+1} \leftarrow \mathbf{w}_k^t - \eta \nabla_{\mathbf{w}} \ell(\mathbf{w}_k^t; D_k),$$

where $\eta$ is the learning rate, and $\ell$ denotes the local loss function. After local updates, the selected clients $K_t$ upload their models to the server. The server then aggregates the updates by computing a weighted average:

$$\mathbf{w}^{t+1} = \sum_{k \in K_t} \frac{|D_k|}{\sum_{k \in K_t} |D_k|} \mathbf{w}_k^{t+1}.$$

## 3.2 TRAINING GENERATOR

Leveraging generators for produce data knowledge distillation is not a novel concept. For instance, Zhang et al. (2022b) introduced **FedFTG**, which uses a server-side GAN to simulate synthetic data based on knowledge aggregated from multiple clients. While this approach effectively captures some unique characteristics from each client's data using hard samples, it falls short in fully harnessing diversity. Similarly, **DENSE** (Zhang et al., 2022a) synthesizes data on the server using a GAN trained on ensemble models uploaded from clients. Although this method strives to generate data that accurately represents the client distributions, it faces limitations in fully capturing the nuanced diversity of each client's local data. In contrast, our method avoids reliance on a single centralized generator by employing a group of GAN models, each specifically tailored to its client's data. At this stage, well-behaved generators $G_i$ are trained on each client $i$, capturing the data distribution $D_i$ over the feature space $\mathcal{X}_i$. Instead of uploading compressed representations to the server, we upload the trained GAN models $\{G_i\}_{i=1}^N$, preserving the diversity $\mathcal{D}$ of each client's local data. To validate our approach, we compare the performance of different generator training strategies. Specifically, we assess a single generator $G$ trained on a global dataset $D_{\text{global}}$ against multiple GANs $\{G_i\}$, each trained on highly skewed, non-IID datasets $D_i$. The results in Fig. 2 demonstrate that the data quality remains comparable across both methods, confirming the robustness of our distributed GAN setup in addressing data heterogeneity.

In our approach, the method for generating synthetic data is inspired by **DeGAN** (Addepalli et al., 2019), a data-free knowledge distillation framework. Building on DeGAN, we adopt a three-player adversarial game between the generator $G_i$, a discriminator $T_i$, and a pre-trained classifier $C_i$ on each client $i$. The generator $G_i$ produces samples from a latent space $\mathcal{Z} \sim \mathcal{N}(0, I)$, while the discriminator $T_i$ ensures that the generated samples align with the distribution of the proxy dataset on client $i$. The classifier $C_i$, a standard model trained on the client, ensures that the generated samples are representative of the true data distribution $D_i$ by minimizing classification entropy.

The generator's loss $L_G$ incorporates three key components. We consider $y$ as the classifier output corresponding to the generator input $z$, where $z$ is sampled from a Gaussian distribution $\mathcal{Z} \sim \mathcal{N}(0, I)$. The expectation over classifier outputs across a batch of samples from the latent space is denoted by $w$:

$$y = C_i(G_i(z)), \quad w = \mathbb{E}_{z \sim \mathcal{Z}}[C_i(G_i(z))]$$

The losses used to train the generator are as follows:

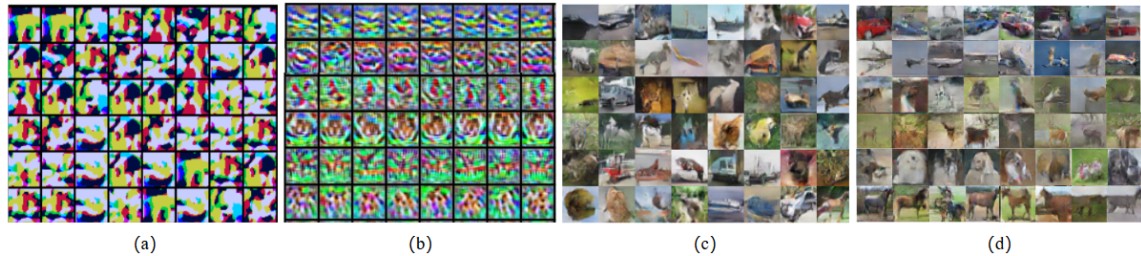

(a)        (b)        (c)        (d)

Figure 2: Illustration comparing various GAN training approaches in identical non-IID data settings with $\omega = 0.01$: (a) FedFTG, where a single generator aggregates knowledge from multiple clients; (b) DENSE, using an ensemble of models from clients to train a centralized generator; (c) a DeGAN-based generator $G$ trained on a global dataset $D_{\text{global}}$; (d) multiple DeGAN-based generators $\{G_i\}_{i=1}^N$ trained on non-IID datasets $D_i$ from different clients. This comparison demonstrates that utilizing a group of GAN models, each tailored to its client's dataset, results in great data quality and diversity.

The adversarial losses (Goodfellow et al., 2014), $L_{\text{adv,real}}$ and $L_{\text{adv,fake}}$, ensure that the distribution of the generated images closely approximates the target data distribution:

$$L_{\text{adv,real}} = \mathbb{E}_{x \sim D_i(x)}[\log T_i(x)], \quad L_{\text{adv,fake}} = \mathbb{E}_{z \sim \mathcal{Z}}[\log(1 - T_i(G_i(z)))]$$

The entropy loss $L_{\text{entropy}}$ reduces the classifier's output uncertainty, ensuring that each generated sample is confidently assigned to one of the classifier's classes:

$$L_{\text{entropy}} = \mathbb{E}_{z \sim Z}\left[-\sum_{k=0}^{C} y_k \log(y_k)\right]$$

where $y_k$ represents the classifier's output for class $k$.

The diversity loss $L_{\text{diversity}}$ ensures that the classifier's outputs across a batch are uniformly distributed among classes, preventing the generated samples from being biased toward any particular class:

$$L_{\text{diversity}} = -\sum_{k=0}^{C} w_k \log(w_k)$$

where $w_k$ is the expected classifier output for class $k$ across the batch.

Building upon the DeGAN framework, we introduce further enhancements to address non-IID data by incorporating an inversion loss, inspired by **DeepInversion** (Yin et al., 2020). This loss $L_{\text{inversion}}$ guides the generator to align the generated data's features with those of the global model. It achieves this by minimizing the discrepancy between the feature statistics of the global model and the generated data, which is formulated as:

$$L_{\text{inversion}} = \sum_{l=1}^{L} \left( \|\mu_l(x) - \mu_l(G(z))\|_2^2 + \|\sigma_l(x) - \sigma_l(G(z))\|_2^2 \right),$$

where $\mu_l(x)$ and $\sigma_l(x)$ represent the running mean and variance of the feature maps at layer $l$ in the global model, while $G(z)$ denotes the generator's output. By focusing on these feature statistics, the inversion loss pushes the generator towards learning representations consistent with the global model's feature space.

The sign of the hyperparameter $\lambda_{\text{inv}}$ plays a crucial role in controlling the behavior of the generator. When $\lambda_{\text{inv}}$ is positive, it works in coordination with the diversity loss to enhance the variety of the generated

samples, encouraging a broader range of features to be represented by integrating information from multiple data distributions. Conversely, a negative $\lambda_{\text{inv}}$ shifts the focus toward local data specifics, allowing the generator to capture the unique aspects of the local data distribution and produce more specialized samples.

The generator's loss $L_G$ builds upon the adversarial, entropy, and diversity components introduced in De-GAN, with the inversion loss added to adapt to non-IID data. The total loss is expressed as:

$$L_G = L_{\text{adv}} + \lambda_e L_{\text{entropy}} - \lambda_d L_{\text{diversity}} + \lambda_{\text{inv}} L_{\text{inversion}},$$

where $\lambda_e$, $\lambda_d$, and $\lambda_{\text{inv}}$ are hyperparameters that control the relative importance of the entropy, diversity, and inversion losses, respectively.

### 3.3 ENSEMBLE DISTILLATION

Rather than aggregating models solely by sample quantities (Qi et al., 2024), we propose an approach that capitalizes on task-specific data distributions to form an ensemble. Specifically, we aggregate models from $N$ clients according to the distribution of class-specific labels, resulting in $C$ specialized models, each dedicated to a particular class. The aggregation for class $c$ is formalized as:

$$w_c^{(t+1)} = \sum_{i=1}^{N} \frac{|D_{c,i}|}{|D_{c,\text{total}}|} w_i^{(t)},$$

where $w_c^{(t+1)}$ represents the aggregated model for class $c$ at round $t+1$, $|D_{c,i}|$ is the number of samples of class $c$ held by client $i$, and $|D_{c,\text{total}}| = \sum_{i=1}^{N} |D_{c,i}|$ is the total number of samples of class $c$ across all clients.

By aggregating $C$ specialized models, the ensemble exploits the individual strengths of each model, better addressing the heterogeneity of data distributions than a single global model. Once the ensemble is established, an attention-based meta-head $M$ is introduced to dynamically adjust the weights $\alpha_c$ for each model within the ensemble. This meta-head, built upon a transformer architecture (Vaswani et al., 2017), ensures that the ensemble achieves optimal performance across tasks.

In the proposed meta-training framework, each meta-training cycle consists of multiple rounds, denoted by $t$, in which the server selects a subset of clients $K_t$ to receive the ensemble model $\mathcal{E}^{(t)}$ and the meta-head $M^{(t)}$. Notably, while both the ensemble and the meta-head are distributed to the clients, only the updated meta-head $M^{(t+1)}$ is uploaded to the server for aggregation after local training, with the ensemble model $\mathcal{E}^{(t)}$ kept frozen throughout the entire meta-training process. During each round $t$, clients refine the meta-head $M^{(t)}$ using their local datasets $D_k$, aggregating the predictions of each model within the ensemble as follows:

$$y_{\text{meta},k} = \sum_{c=1}^{C} \alpha_c^{(t)} y_{c,k},$$

where $y_{c,k}$ represents the prediction of each model in the ensemble for client $k$, and $\alpha_c^{(t)}$ are the corresponding weights learned by the meta-head at cycle $t$. This process is repeated across multiple rounds within a meta-training cycle, typically spanning $T$ rounds.

An Exponential Moving Average (EMA) (Kingma & Ba, 2014) is applied to the meta-head, stabilizing the training process and mitigating catastrophic forgetting. The EMA update is expressed as:

$$\alpha_c^{(t+1)} = \beta \alpha_c^{(t)} + (1 - \beta) \alpha_c^{(t+1)},$$

where $\beta$ is the decay rate, controlling how much of the previous meta-head weights are retained during each update. This process unfolds over several cycles, allowing the meta-head to steadily enhance its performance.

To further augment the global model, we employ the synthetic dataset $\{(X_i^S, Y_i^S)\}_{i=1}^N$ generated by the GAN group $\{G_i\}_{i=1}^N$, where $X_i^S$ represents the generated data samples and $Y_i^S$ are the corresponding labels produced by the ensemble. This data is then leveraged to distill knowledge from the ensemble of specialized models into the global model, serving as a student. This data-free knowledge distillation enhances the global model's ability to generalize across all classes, thus improving performance in non-IID scenarios.

## 4 EXPERIMENTS

### 4.1 EXPERIMENTAL SETUP

**Datasets**. In this study, we assess the performance of various methods using two image classification datasets, CIFAR-10 and CIFAR-100 (Alex, 2009). To simulate the inherent data heterogeneity among clients, we follow the approach adopted in previous works (Luo et al., 2023; Wang et al., 2020; Yurochkin et al., 2019), where the Dirichlet distribution $\text{Dir}(\omega)$ is applied to partition the training dataset for each client. The concentration parameter $\omega$ controls the extent of data heterogeneity, with smaller values of $\omega$ resulting in more non-uniform data distributions. The same partitioning process is employed for both CIFAR-10 and CIFAR-100 datasets. This setup provides a suitable foundation for evaluating the effectiveness of our proposed methods under different levels of data non-IID conditions.

**Baselines**. We compare our method with the following baselines: FedAvg (McMahan et al., 2016), FedRS (Li & Zhan, 2021), Focal Loss (Lin et al., 2017), FedLF (Lu et al., 2024), DENSE (Zhang et al., 2022a), DFRD (Luo et al., 2023), and FedFTG (Zhang et al., 2022c). The first four methods focus on addressing data heterogeneity, while the last three methods, similar to ours, are based on data-free knowledge distillation techniques. These methods extract knowledge from local models at the client side to synthesize data and perform knowledge distillation on the global model in a fine-tuning manner. We place particular emphasis on comparing the performance of these latter three approaches. Further configurations can be found in **Appendix A.2**.

### 4.2 RESULTS AND ANALYSIS

We conducted an in-depth analysis of the performance of various methods under different degrees of data heterogeneity on the CIFAR-10 and CIFAR-100 datasets, as shown in Table 1. In the table, **bold** results represent the highest accuracy, and underlined results represent the second-highest accuracy for the global model in each column. It is evident that as the value of $\omega$ decreases, all methods experience a significant performance degradation. Our proposed method, DFED, consistently outperforms the baseline method, FedAvg, across various settings. The first four methods listed in the table—FedAvg, FedRS, FedLF, and LocalLoss—are not data-free knowledge distillation approaches, yet they still demonstrate robust capabilities in handling data heterogeneity. In contrast, the latter three methods—FedFTG, DFRD, and Dense—are data-free knowledge distillation methods, which serve as the primary focus of our comparative analysis. Further analysis and discussions can be found in **Appendix A.3**.

### 4.3 ABLATION STUDY

In this section, we rigorously demonstrate the efficacy and indispensability of the core modules and key hyperparameters of our method under the same settings. To assess their impact, particularly the inversion loss during GAN training process and the meta-head in ensemble learning, we conduct a series of ablation experiments. By systematically removing or adjusting these elements, we aim to discern their individual contributions to the model's performance. Further analysis and discussions can be found in **Appendix A.4**.

Table 1: Top test accuracy (%) of distinct methods across $\omega \in \{0.01, 0.1, 1.0\}$ on CIFAR-10 and CIFAR-100 datasets.

| Algs. | CIFAR-10 | | | CIFAR-100 | | |
|---|---|---|---|---|---|---|
| | $\omega = 1.0$ | $\omega = 0.1$ | $\omega = 0.01$ | $\omega = 1.0$ | $\omega = 0.1$ | $\omega = 0.01$ |
| FedAvg | $69.18_{\pm 1.10}$ | $54.31_{\pm 1.83}$ | $32.41_{\pm 2.75}$ | $44.16_{\pm 0.37}$ | $38.56_{\pm 0.51}$ | $29.71_{\pm 1.38}$ |
| FedRS | $76.62_{\pm 1.23}$ | $\mathbf{70.14}_{\pm 1.65}$ | $34.24_{\pm 1.97}$ | $\underline{50.17}_{\pm 0.48}$ | $41.02_{\pm 0.65}$ | $31.29_{\pm 1.04}$ |
| FedLF | $\mathbf{79.63}_{\pm 1.80}$ | $\underline{69.21}_{\pm 1.59}$ | $32.84_{\pm 1.42}$ | $\mathbf{53.10}_{\pm 0.36}$ | $43.37_{\pm 0.28}$ | $32.77_{\pm 1.24}$ |
| Focalloss | $\underline{76.64}_{\pm 1.67}$ | $66.83_{\pm 1.22}$ | $34.41_{\pm 2.13}$ | $46.11_{\pm 0.71}$ | $36.27_{\pm 0.33}$ | $29.64_{\pm 1.08}$ |
| FedFTG | $69.88_{\pm 1.26}$ | $56.27_{\pm 1.62}$ | $35.71_{\pm 1.69}$ | $45.41_{\pm 0.23}$ | $39.82_{\pm 0.49}$ | $30.31_{\pm 1.46}$ |
| DFRD | $72.03_{\pm 0.91}$ | $59.74_{\pm 1.21}$ | $\underline{40.42}_{\pm 1.65}$ | $49.45_{\pm 0.27}$ | $\underline{43.49}_{\pm 0.99}$ | $\underline{33.28}_{\pm 1.18}$ |
| DENSE | $69.73_{\pm 0.69}$ | $55.49_{\pm 1.16}$ | $33.85_{\pm 1.22}$ | $45.41_{\pm 0.35}$ | $39.25_{\pm 0.82}$ | $30.54_{\pm 1.55}$ |
| DFED | $71.27_{\pm 0.94}$ | $60.15_{\pm 1.11}$ | $\mathbf{42.17}_{\pm 1.83}$ | $48.89_{\pm 0.33}$ | $\mathbf{44.11}_{\pm 0.67}$ | $\mathbf{34.28}_{\pm 1.99}$ |

Table 2: Comparison of Different Ensemble Methods on CIFAR-10 Dataset Across Various $\omega$ Values.

| Ensemble Method | CIFAR-10 | | |
|---|---|---|---|
| | $\omega = 1.0$ | $\omega = 0.1$ | $\omega = 0.01$ |
| DENSE-ensemble | $62.22_{\pm 2.69}$ | $50.15_{\pm 2.13}$ | $24.95_{\pm 3.32}$ |
| DFED-ensemble-basic | $77.64_{\pm 1.33}$ | $59.21_{\pm 1.89}$ | $40.41_{\pm 0.98}$ |
| DFED-ensemble-meta | $79.09_{\pm 0.45}$ | $63.15_{\pm 1.11}$ | $54.33_{\pm 1.12}$ |
| DFED-ensemble-meta-EMA | $80.12_{\pm 0.84}$ | $65.44_{\pm 0.76}$ | $59.86_{\pm 1.70}$ |

## 5 CONCLUSION

In this work, we present a novel federated learning framework designed to improve model performance in heterogeneous environments. Our approach utilizes GANs at the client level to handle data imbalance, where each client trains its own GAN, generating diverse synthetic data while maintaining privacy and ensuring unique distribution characteristics. By integrating model ensembles with attention-based meta-learning, we significantly enhance the ensemble's performance, surpassing traditional global models. Furthermore, we employ knowledge distillation using both the synthetic data generated by the GANs and the high-performing ensemble, leading to further improvements in accuracy. Our method achieves superior results compared to several state-of-the-art baselines, as demonstrated on the CIFAR-10 and CIFAR-100 datasets.

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

# A APPENDIX

## A.1 RELATED WORK

Heterogeneous federated learning (HFL) has emerged as a crucial field of study, primarily due to the diverse and decentralized nature of client environments and data distributions(Gao et al., 2022). One of the central challenges in HFL is addressing data heterogeneity and ensuring robust performance across non-IID data distributions. To address these issues, Xu et al. (2021) develop an adaptive federated averaging technique that enhances communication efficiency and reduces convergence time by dynamically adjusting learning rates to better accommodate local data distributions. Additionally, Tan et al. (2023) propose FedICON, which uses contrastive learning to handle feature shifts by extracting invariant information across clients, enhancing robustness in non-IID federated learning scenarios. In parallel, Shen et al. (2023) propose a closed-form classifier framework that enhances cross-device learning by optimizing aggregation strategies, resulting in faster convergence and more stable training dynamics. While these methods offer substantial advancements, they often neglect the challenge of client drift, a phenomenon where the non-IID nature of data causes divergence in client updates, leading to misaligned aggregation. This drift impairs the global model's ability to converge effectively. As a result, without adequately addressing client drift, existing approaches may struggle to maintain stability and consistent performance as data heterogeneity increases in federated learning environments.

Data-Free Knowledge Distillation (DFKD) has become a pivotal approach in scenarios where data privacy and availability are constrained. In contrast to traditional distillation methods that require access to original training data, DFKD facilitates knowledge transfer from teacher to student models by generating synthetic data, ensuring the protection of sensitive information. Recent advancements in this domain have introduced innovative techniques aimed at improving the quality and efficiency of synthetic data generation. For instance, Yu et al. (2023) employ channel-wise feature exchange and spatial activation region constraints to enhance data diversity, resulting in more robust student models without relying on real data. Similarly, Tran et al. (2023) propose NAYER, a method that shifts the source of randomness to a noisy layer, paired with label-text embeddings to produce high-quality samples. This approach accelerates the training process while maintaining competitive accuracy. Another significant contribution comes from Shin & Choi (2024), who present the Teacher-Agnostic DFKD (TA-DFKD), which redefines the role of the teacher model as a lenient expert, allowing for more diverse sample generation by reducing class-prior restrictions. Despite these innovations, DFKD still faces challenges in generating diverse, high-fidelity samples. Methods often struggle to capture the full distribution of the original data, especially in imbalanced scenarios, which can lead to biased student models. Nonetheless, DFKD continues to evolve, driven by the increasing demand for privacy-preserving techniques in machine learning, establishing itself as a rapidly advancing field.

Data-Free Knowledge Distillation (DFKD) in Federated Learning (FL) offers a privacy-preserving solution for knowledge transfer, eliminating the need for raw data exchanges between clients. By generating synthetic data for distillation, DFKD ensures sensitive information remains protected while facilitating effective knowledge transfer from global teacher models to local student models. This approach is particularly suitable for handling data heterogeneity and non-IID distributions, as these issues often undermine model aggregation in FL. Luo et al. (2023) introduce DFRD, a method that employs a conditional generator on the server to synthesize training data, addressing distribution shifts and enhancing the diversity of synthetic samples. Yang

et al. (2023) propose FedFed, a framework designed to combat data heterogeneity through feature distillation. In this method, clients retain robust features locally while sharing performance-sensitive features with added noise, significantly improving model performance without compromising privacy. Similarly, Zhang et al. (2024a) present FedKTL, a knowledge transfer method that leverages a server-side pre-trained generator, efficiently addressing both model and data heterogeneity while minimizing communication overhead. While these methods excel in generating diverse synthetic data and have demonstrated impressive effectiveness in addressing data heterogeneity through DFKD, they fall short in mitigating client drift, which can lead to misaligned updates in non-IID settings. Our approach, by employing an ensemble learning strategy, not only preserves data diversity but also effectively tackles client drift. This ensures greater stability and enhanced performance in federated learning environments, offering a more comprehensive solution to both data diversity and alignment challenges.

## A.2 Experimental setup

**Configurations**. Unless otherwise specified, all experiments are conducted in a centralized network with $N = 10$ active clients. To simulate varying degrees of data heterogeneity, we use $\omega \in \{0.01, 0.1, 1.0\}$, where smaller values of $\omega$ indicate stronger data imbalances. All baselines adopt the same configuration to ensure fair comparison. All experiments utilize ResNet-18 (He et al., 2016) as the base model and are executed in PyTorch on an Nvidia GeForce RTX 3080 GPU. Unless stated otherwise, most hyperparameters for these baselines are configured according to the original literature, and we utilize the official open-source codes for these methods. Regarding the meta-training process, we opt to update the meta model every ten communication rounds, setting the meta phase $T$ to 20, with the number of selected clients per round ranging from 1 to 3, contingent upon the distribution setup.

**Evaluation Metrics.** We evaluate the performance of different FL methods solely based on global test accuracy. Specifically, we employ the global model on the server to assess the overall performance of various FL methods using the original test set. To ensure reliability, we report the average results for each experiment over 5 different random seeds.

## A.3 Analysis in our Experiments

When $\omega = 1$, the data distribution across clients is relatively uniform. Although data-free knowledge distillation methods can address data heterogeneity to some extent, they fail to exhibit a significant advantage in this scenario, as the knowledge disparity between clients is not sufficiently pronounced. However, as $\omega$ decreases to 0.01, exacerbating the data heterogeneity, the advantages of data-free knowledge distillation become more pronounced. In this extreme scenario, DFED achieves the best overall performance, demonstrating its superior ability to handle highly heterogeneous data environments. Notably, both DENSE and DFED leverage ensemble methods in their respective frameworks. The results of the comparison are presented in Table 2. In our data partitioning experiments, we evaluated the performance of DENSE's ensemble strategy; however, its ensemble yielded lower accuracy compared to the global model. This outcome can be attributed to DENSE's simplistic approach of averaging the outputs of the client models, which does not necessarily yield optimal results as it may fail to effectively account for the specialized strengths of individual client models based on their specific expertise. In contrast, our ensemble method, applied under the same partitioning scheme, achieved remarkable performance across a variety of configurations, significantly surpassing the results of the DENSE ensemble.

Overall, our method achieved impressive outcomes in all experiments. Although it performed slightly below the first four methods when data heterogeneity was less pronounced, it surpassed the three data-free knowledge distillation methods. Furthermore, our approach yielded exceptional results under extreme partitioning conditions.

A.4 ABLATION STUDY

**Impacts of hyperparameters on the GAN group's loss components.** Building upon the DeGAN framework, we include the adversarial loss $L_{adv}$, entropy loss $L_{entropy}$, and diversity loss $L_{diversity}$, with the additional inversion loss $L_{inv}$ incorporated to handle the challenges posed by non-IID data distributions. Our primary focus is on the inversion loss. We observe that the quality of data generated by the GAN group is influenced by the number of clients participating in training process. When a small proportion of clients participate in the training, the inversion loss significantly enhances the quality of the generated data. However, as the majority of clients are involved, the inversion loss diminishes its effectiveness and, at larger scales, begins to hinder the overall data generation process. When the inversion loss is negative, it introduces considerable instability, generally resulting in adverse effects on the training dynamics and overall model performance. However, when using a ResNet18 classifier trained on the homogeneous dataset, such as CIFAR-10 with a classifier pre-trained on CIFAR-100, the negative inversion loss contributes to performance improvement. We found that setting the hyperparameter $\lambda_{inv}$ to 10 is most suitable, and it is preferable to omit the inversion loss when the number of active clients exceeds 60%, while applying inversion loss is more beneficial when the number of active clients is below 60%.

**Impacts of the meta-head on ensemble learning.**

Our approach aggregates models according to the local data distributions of each client, resulting in improved accuracy by leveraging models that specialize in specific data categories. Subsequently, we leverage a transformer-based meta-head to assign adaptive weights to the outputs of the model ensemble. During meta-training, we select and distribute 1 client model per round, updating the meta-head after each round. In our configuration, 50 rounds of meta-training strike a balance between communication overhead and training adequacy, as more rounds increase communication costs, while fewer rounds may lead to underfitting. In the case of CIFAR-100, which contains a larger number of categories, we distribute the training weights for only a subset of the ensemble models per round, rather than distributing all 100 models at once. Table 2 presents the results. The term "basic" refers to the models that were not trained using the meta-head, while "meta" indicates that the outputs were weighted using the meta-head during training without applying EMA. In contrast, the "meta-EMA" column represents the results where EMA was applied to the meta-head during training to further stabilize the model.

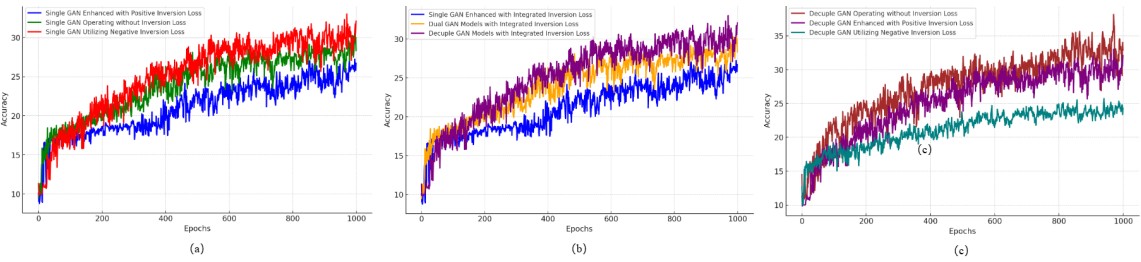

Figure 3: Illustration of global model accuracy curves during the knowledge distillation process using GAN-generated images on the CIFAR-10 dataset. (a) Comparison of a single GAN model trained with positive inversion loss, without inversion loss, and with negative inversion loss using a classifier pre-trained on the CIFAR-100 dataset. (b) Comparison of the number of GANs using positive inversion loss. (c) Comparison of ten models utilizing different loss configurations.

### A.5 POTENTIAL ERRORS IN COMPARATIVE EXPERIMENTS

In the interest of transparency, we must disclose some potential issues encountered during the comparative experiments. We referred to the open-source codebases of FedLF (Lu et al., 2024) and DFRD (Luo et al., 2023) for reproduction and comparison, but significant discrepancies were observed. Our experiments were primarily based on the DFRD framework, which includes FedAvg, FedFTG, DENSE, and DFRD itself. Initially, we directly used their provided code for testing; however, the latter three algorithms (FedFTG, DENSE, and DFRD) demonstrated issues on the CIFAR-10 and CIFAR-100 datasets, where the global model's accuracy remained consistently low and failed to converge. Subsequently, we referred to the source code of each method and conducted our own reproduction, which resulted in improved but still varied performance.

For the FedLF codebase, we utilized only FedRS, LocalLoss, and FedLF methods, but observed some inaccuracies and instability. Specifically, the results obtained by applying the Dirichlet-based partitioning method from DFRD to FedLF's open-source code yielded exceptionally strong outcomes, far surpassing the baseline FedAvg. To further investigate the issue, we attempted to reproduce the algorithms within the DFRD framework, and the results were found to be slightly inferior compared to those obtained from the FedLF implementation.

To ensure fairness and respect, we have chosen to present the results obtained using FedLF's open-source code along with our data partitioning method. It is important to note that while there was a substantial performance gap between the methods when $\omega = 1$ and $\omega = 0.1$, the results were consistent in highlighting data heterogeneity issues when $\omega = 0.01$. Due to time constraints, we have not yet fully integrated both codebases, but we aim to provide a more thorough and scientifically rigorous comparison in future open-source releases.

### A.6 ADDITIONAL ANALYSIS ON HYPERPARAMETER IMPACT

In this subsection, we provide further analysis and discussions on the impact of the key hyperparameters introduced in the main text. Building upon the DeGAN framework, we include the adversarial loss $L_{\text{adv}}$, entropy loss $L_{\text{entropy}}$, and diversity loss $L_{\text{diversity}}$, with the additional inversion loss $L_{\text{inv}}$ incorporated to handle the challenges posed by non-IID data distributions. Our experiments reveal a certain degree of homogeneity between the inversion and diversity losses. The integration of global model features facilitates the GAN's ability to generate diverse distributions. However, when the model is exposed to a dataset containing only a single class, the diversity loss fails to assist the generator in synthesizing high-confidence images from other classes, while the inversion loss can partially mitigate this limitation. It is important to highlight that the inversion loss interferes with the discriminator $T$, affecting its confidence in generated samples. Although the generator continues to produce images that exhibit favorable knowledge distillation effects, with most generated samples closely approximating the local data, the discriminator assigns these samples an exceptionally low confidence score, interpreting them as significantly different from the real data. Consequently, in scenarios where only a subset of active clients participate in the federated learning process, the inversion loss aids the GAN group in capturing global information, enabling the generation of richer and more diverse samples. However, when the majority of clients are involved in the training process, the GAN group already possesses a broad range of sample knowledge, reducing the effectiveness of the inversion loss, which may even hinder the synthesis of high-quality samples. Another comparison arises when the inversion loss is set to a negative value, meaning that the generated images are more deviated from the global features and may lean toward specific categories in the local dataset. Additionally, this approach introduces a level of antagonism with the diversity loss. GAN training becomes highly unstable under these conditions, as global features still encompass characteristics of the local data, and in the worst-case scenario, the generated images tend to resemble noise. However, in some of our experiments, the GAN trained with a negative inversion loss outperformed the one trained without inversion loss, particularly for clients that rarely participate in

the training process. We test the negative inversion loss by utilizing a CIFAR-100 classifier on the CIFAR-10 dataset, achieving promising results with a small number of GANs. However, when using a CIFAR-10 classifier on the CIFAR-100 dataset, the results are not as significant.

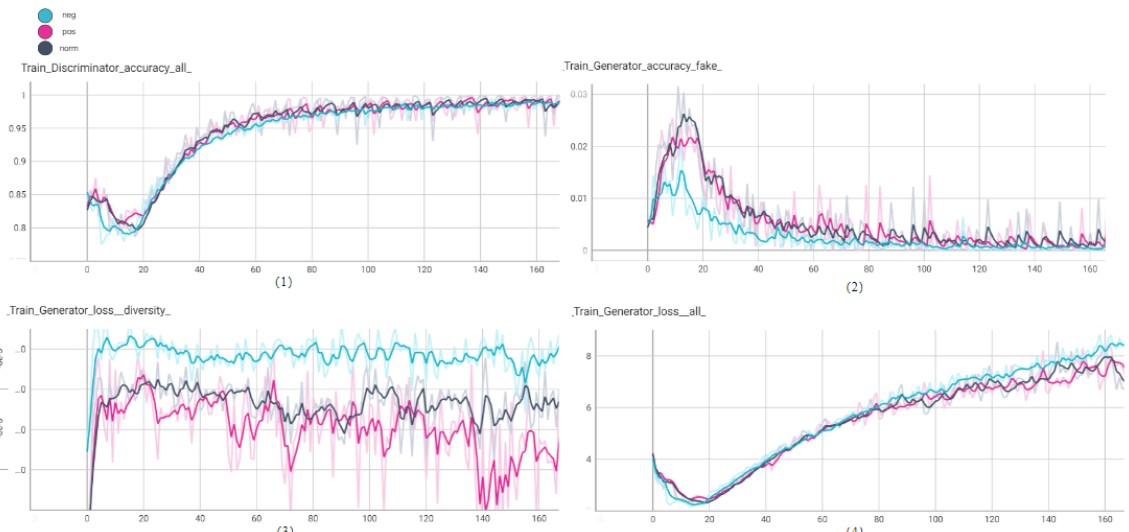

Figure 4: This figure shows an example of the comparison between positive and negative inversion loss and the absence of inversion loss during the GAN training process.

We also conducted experiments with varying numbers of GAN models and different values of the inversion loss hyperparameter, $\lambda_{inv}$. Our findings indicate that excessively high or low values of $\lambda_{inv}$ negatively impact performance, diminishing the quality of both the discriminator and the generator, ultimately affecting the efficacy of knowledge distillation.

## A.7 LIMITATIONS AND SHORTCOMINGS OF OUR METHOD

The foremost limitation of our method is the substantial communication overhead it generates, as well as the high storage requirements for the clients. This is evident in several aspects: in terms of communication, both the GANs and the local models are uploaded to the server, and during the meta-training phase, an ensemble of models is distributed to clients for multiple rounds of communication. This results in a considerable communication burden, which may not be feasible in practical applications. While this is still manageable for the CIFAR-10 dataset, the ensemble for CIFAR-100 becomes too large. To mitigate this, we distribute a subset of the ensemble models for weight updates in each round, rather than all 100 models at once, thereby reducing the communication load across more rounds. However, this also means that the typical federated learning training process will be paused for an extended period during these rounds.

In terms of storage, we assume that the server has unlimited storage capacity, but for clients, it is challenging to store large-scale models and provide sufficient memory for training. This presents a significant limitation of our method in practical applications. Our approach essentially trades off space and time for better performance, which is a key aspect of our design philosophy.

Another shortcoming of our method lies in the DeGAN framework. We have not conducted in-depth research on this data synthesis technique and have borrowed methods from other works, which may not be fully suited to our use case. In both DeGAN and traditional data-free knowledge distillation methods, the teacher model

typically has very high accuracy. For instance, 95% of the teacher models have excellent features, enabling the training of student models with up to 80% accuracy on the CIFAR datasets. However, when the accuracy of the teacher model drops to around 60%-80%, the effectiveness of knowledge distillation is significantly reduced. Our 60% model ensemble can only distill student models with around 40% accuracy, and the 80% model ensemble can only distill student models with approximately 60% accuracy, which represents a major loss in efficiency.

If we had access to a public dataset, the accuracy of the student model post-distillation could even surpass that of the model ensemble. We conducted some preliminary experiments on the CIFAR-10 dataset to test this hypothesis.

