# OpenReview forum: "DFED: Data-Free Ensemble Distillation with Multi-Source GANs for Heterogeneous Federated Learning"
_ICLR.cc/2025/Conference — ICLR 2025 Conference Withdrawn Submission_

### Official Review · Reviewer_kpbK · 2024-10-28

**Soundness:** 2
**Presentation:** 3
**Contribution:** 1
**Rating:** 3
**Confidence:** 5

**Summary:**

This paper focuses on heterogeneity in federated learning. It proposes a data-free ensemble knowledge distillation method that leverages multi-source GANs to generate synthetic data aligned with local distributions. Moreover, the proposed method aggregates client models into an ensemble and refines the global model by ensemble distillation. Abundant experiment results demonstrate the effectiveness of the proposed method.

**Strengths:**

1. The paper is easy to follow.
2. The proposed method shows robustness under various data heterogeneous scenarios.

**Weaknesses:**

1. My primary concern is that uploading GAN groups would result in data leakage.
2. The proposed ensemble distillation method contains C specialized models, what is the computation burden of the proposed method? What if C is too large?
3. How is the scalability of the proposed method? For example, different numbers of clients.
4. The paper devotes more than a page to introducing the GAN part, which is essentially a tailored module from DeGAN and DeepInversion.
5. The performance of the proposed method is not compelling. It only archives SOTA performance in 3 of 6 settings,
6. Experiments on more challenging datasets are required, like tiny imagenet.

**Questions:**

Please check the weakness part for details.

---

### Official Review · Reviewer_RnbZ · 2024-10-31

**Soundness:** 3
**Presentation:** 3
**Contribution:** 2
**Rating:** 3
**Confidence:** 4

**Summary:**

The paper proposes DFED—a novel approach using data-free knowledge distillation (DFKD) and GANs to generate synthetic, diverse client data. DFED combines GANs for data generation, a model ensemble for task-specific focus, and attention-based meta-learning to enhance performance, significantly outperforming existing methods on CIFAR-10 and CIFAR-100 datasets.

**Strengths:**

1. The proposed method effectively alleviates the data heterogeneity issue in federated learning.
2. The use of GANs is straightforward and effective, providing a privacy-preserving and performance-enhancing approach.
3. The experiments rigorously demonstrate the method's effectiveness, but I believe some special cases still need consideration.
4. The writing throughout is fluent, allowing me to understand both the overall approach and detailed aspects of the method well.

**Weaknesses:**

1. The comparative experiments in the paper are insufficient, as many SOTA methods, such as FedInit [1] and FedOPT [2], have not been included for comparison. Explain the criteria for selecting baseline methods and include a brief discussion on why certain SOTA methods like FedInit and FedOPT were not included, and how their inclusion might affect the comparative analysis.

2. Requiring each client to provide an additional GAN imposes a significant storage burden, and the resulting minor performance gain does not seem meaningful to me. Provide a quantitative analysis of the storage requirements for the GANs versus the performance gains achieved. This could include a breakdown of storage costs per client and a comparison of performance improvements across different dataset sizes or heterogeneity levels.

3. I am skeptical that the GAN can accurately learn the data distribution of a client when the client’s data volume is very limited. Please include an analysis of the GAN's performance under different client data volume scenarios. Provide empirical results or theoretical bounds on the minimum data volume required for the GAN to learn an accurate distribution.

4. Providing the GAN to the server could increase the risk of privacy leakage. Provide a detailed privacy analysis of their method. Please discuss potential privacy risks associated with sharing GANs, compare these risks to existing federated learning approaches, and propose any mitigation strategies they have considered.

[1]. Sun Y, Shen L, Tao D. Understanding how consistency works in federated learning via stage-wise relaxed initialization[J]. Advances in Neural Information Processing Systems, 2023, 36: 80543-80574.
[2]. Reddi S, Charles Z, Zaheer M, et al. Adaptive federated optimization[J]. arXiv preprint arXiv:2003.00295, 2020.

**Questions:**

see weakness

---

### Official Review · Reviewer_cz39 · 2024-11-03

**Soundness:** 2
**Presentation:** 2
**Contribution:** 1
**Rating:** 3
**Confidence:** 5

**Summary:**

This paper proposes a novel FL method called DFED, which leverages multi-source Generative Adversarial Networks (GANs) to generate synthetic data that aligns with local distributions, ensuring privacy while promoting diverse feature representations across clients. Additionally, DFED aggregates client models into an ensemble based on their specialized knowledge, and applies ensemble distillation to refine the global model, mitigating the issues caused by disparities in data distributions. Empirical experiments with extensive analysis on image classification datasets demonstrate the superiority of DFED in terms of test accuracy.

**Strengths:**

1. The paper is well-written and easy to follow.

2. Data-free black-box knowledge transfer across heterogeneous clients in Federated Learning (FL) is interesting and promising.

**Weaknesses:**

1. This study requires more communication costs compared to existing methods such as FedAvg, DFRD, and DENSE.

2. This study lacks innovation, as previous studies have used similar simpler strategies but have a wider range of applicability, such as FL for model heterogeneity.

3. Lack of experiments on larger models and datasets, as well as some important ablation experiments.

**Questions:**

Details:
1. From Figure 1, DFED can be seen that the clients download the global model and Meta from the server, while the clients upload the local model, Meta, and GANs to the server. This results in a significant amount of additional communication costs, making it difficult to meet the bandwidth limitations of a large number of FL scenarios.

2. I have doubts about the innovation of this work. I argue that using data-free black-box knowledge transfer across heterogeneous clients is interesting and promising. However, this work not only requires the client to upload a local model, but also to upload a GAN trained on local private data. This increases the risk of privacy breaches compared to existing methods DENSE and DFRD. In addition, existing work[1] seems to use a similar strategy (using variational autoencoder). More importantly, DFED seems to fail in FL scenarios with heterogeneous models.

3. The experimental only consider federated learning scenarios with 10 clients. It is recommended to conduct experiments on a larger number of clients, such as K=50 or 100, etc.

4. All the report results in the paper are the final evaluation indicators, such as accuracy, which is insufficient. Therefore, the learning curves and communication rounds should also be reported to demonstrate the training process of DFED, like DFRD.

5. The datasets used in the experimental section are insufficient. It is recommended to conduct experiments on the more difficult SVHN and Tiny ImageNet datasets, while considering larger models.

6. From the method description section, it can be inferred that multiple hyperparameters such as $\lambda_e$, $\lambda_d$, $\lambda_{inv}$ and $\beta$ are introduced during the training process of DFED. However, the ablation experiment lacks detailed numerical experimental research on them, and simple discussions are not enough.

[1] Heinbaugh C E, Luz-Ricca E, Shao H. Data-free one-shot federated learning under very high statistical heterogeneity[C]//The Eleventh International Conference on Learning Representations. 2023.

---

### Official Review · Reviewer_UgQe · 2024-11-05

**Soundness:** 3
**Presentation:** 3
**Contribution:** 2
**Rating:** 5
**Confidence:** 4

**Summary:**

This paper presents an innovative federated learning approach called DFED that improves model effectiveness in heterogeneous environments. First, DFED addresses data imbalance issues by incorporating GANs, with each client training its own GAN to generate synthetic data. This decentralized approach preserves both data uniqueness and privacy. Additionally, DFED leverages a combination of model ensembles and attention-based meta-learning, enabling the ensemble to surpass the performance of a conventional global model. Finally, DFED applies knowledge distillation using the synthetic data generated, further enhancing the high-performing ensemble model.

**Strengths:**

1.The experimental content is relatively comprehensive, and the technical background such as knowledge distillation is clearly introduced.

2.Despite introducing additional computational and communication overhead, optimizing meta head results in better generalization performance of the global model.

3.DFED accurately captures client data features under different distributions using GAN groups.

**Weaknesses:**

1 In the experiment in Table 1, there is already a significant data drift phenomenon at ω=0.1, but the advantage of DFED can not be clearly observed.

2 The communication bottleneck is also one of the challenges in federated learning, as the additional communication and computational overhead introduced in DFED is difficult to ignore. This point is also stated in the appendix.

3 The innovation of gan group or meta head is limited.

**Questions:**

1. The research on diffusion models is becoming increasingly hot. Have you considered using diffusion models as generative models and further optimizing diffusion models in distributed environments?

2. The data in Table 1 and Table 2 seem to be inconsistent.

3. As stated in this paper, the effectiveness of data distillation will vary depending on the ability of the teacher model, which is detrimental to designing a robust data heterogeneous federated framework. Have you considered extending the framework to federated learning with heterogeneous models?

---

### Author Response · Authors · 2024-11-13
**Acknowledgments**

We would like to express our sincere gratitude to each reviewer for their invaluable feedback, and we deeply appreciate the time and effort dedicated by the organizing committee. Given the low likelihood of acceptance, we have decided to withdraw our submission. We will carefully consider each reviewer's suggestions, conduct further experiments, and refine our paper in preparation for future opportunities. Thank you once again for your understanding.

---

### Note · Authors · 2024-11-13

I have read and agree with the venue's withdrawal policy on behalf of myself and my co-authors.